# Landscape Patterns and Topographic Features Affect Seasonal River Water Quality at Catchment and Buffer Scales

**Li Deng** [1], **Wanshu Li** [2,3], **Xiaojie Liu** [2,3,*], **Yazhu Wang** [4] and **Lingqing Wang** [2,3]

1. Ecological Environment Planning and Environmental Protection Technology Center of Qinghai Province, Xining 810007, China
2. Key Laboratory of Natural Resource Coupling Process and Effects, Ministry of Natural Resources, Institute of Geographic Sciences and Natural Resources Research, Chinese Academy of Sciences, Beijing 100101, China
3. University of Chinese Academy of Sciences, Beijing 100049, China
4. Key Laboratory of Watershed Geographic Sciences, Nanjing Institute of Geography and Limnology, Chinese Academy of Sciences, Nanjing 210008, China
* Correspondence: liuxj@igsnrr.ac.cn

**Abstract:** Effects of landscape patterns or topographic features on the river water environment have been broadly studied to control non-point source (NPS) pollution and to cut off potential pathways for pollutants to affect human health. However, spatio-temporal dynamics and scale effects with respect to the impact of landscape patterns and topographic features on the aquatic environment over successive years have not been elucidated. In this study, water quality parameters and land cover data for three consecutive years mainly in Tangshan City, located in the northeast of the Haihe River Basin, China, were obtained to determine the associations between landscape patterns and topographic features with the water environment. Results indicated that seasonal differences in dissolved oxygen (DO) and total nitrogen (TN) were significant ($p < 0.001$), and spatial variation was generally observed for each water quality parameter. Redundancy analysis revealed that landscape patterns and topographic features have different impacts on the aquatic environment as seasonal spans and spatial scales change. Overall, the best explanatory variables explained an average of 58.6% of the variation in water quality at various spatial scales over the two seasons. Topographic features made a greater contribution to river water quality changes at the buffer scale; conversely, at the catchment scale, water quality changes stemmed primarily from differences in landscape composition and configuration. The landscape shape index of cropland ($LSI_{crop}$) was an important factor influencing seasonal river water quality changes at various spatial scales. These results suggest that considering landscape connectivity at distinct spatial scales could enhance the understanding of the alteration of hydrological processes across multiple topographic features, which in turn has an impact on seasonal river water.

**Keywords:** seasonal river water quality; landscape pattern; topographic feature; RDA analysis; Haihe River Basin



## 1. Introduction

Deterioration of water quality is a worldwide problem, exacerbating water scarcity and damaging aquatic ecosystems [1–4]. As a typical river basin affected by human activities, surface water pollution was once a serious problem in the Haihe River basin, located in North China [5]. Water pollution in coastal areas of North China has exacerbated the inconsistency in water allocation and regional development, as well as affecting human life and health [6,7]. In recent years, the implementation of environmental policies has alleviated water pollution. Normally, the apparent spatio-temporal variation in water quality is mainly caused by rainfall, soil erosion, climate perturbation and anthropogenic disturbance [8–15]. Direct pollution of surface water quality by human interventions such as industrial effluent

discharge still needs attention. However, the indirect effects of human activities on water environment, such as the reshaping of landscape patterns, cannot be ignored.

Landscape pattern represents both composition and configuration of landscape elements, generally expressed as the proportion of land use/land cover (LULC) area and the spatial structure of LULC [16,17]. The landscape composition and configuration affect the amount of pollution load and the hydrological process of contaminants transfer into the water [18,19]. As reported in earlier studies, landscape patterns significantly influence water environment variability [20,21]. Cropland and impervious layers facilitate the transport of non-point source (NPS) pollutants and harm rivers [22]. Conversely, forest and grassland, with high vegetation coverage, are useful in controlling the number of contaminants entering the river [23]. Furthermore, LULC configuration, e.g., characterised by patch density, has been demonstrated to be one of the critical factors in regulating water quality as well [24]. However, studies have revealed that landscape patterns have contradictory effects on the aquatic environment at different spatial scales. Studies have demonstrated that landscape patterns at the catchment scale better interpret the variations of aquatic environment [14,25]. Earlier studies have provided the opposite results; the impact of landscape patterns on the aquatic environment was more significant in the buffer zone [26]. Since such variations are likely to originate from differences in geomorphology [27], studies to explore the impact of landscape patterns on the aquatic environment with consideration of geomorphic characteristics have gradually attracted more attention [28].

It is also worth noting the seasonal differences in the impact of landscape patterns on the aquatic environment [29,30]. The study employed partial least squares regression revealing that the contribution of land use composition and structure to specific pollutants (e.g., $NH_3$-N) varied in wet and dry seasons [31]. Explanations for changes in $NH_3$-N were dominated by patch density during the wet season, while the effect of patch density decreased during the dry season. In addition, interannual-scale water quality changes may be influenced by landscape alterations due to human intervention.

Considering several spatial and temporal scales, there is no unified conclusion on the interpretation of the impact of landscape patterns and topography on the aquatic environment. Meanwhile, most studies focus on the total contributions or the contribution of a single landscape pattern indicator to water quality or integrity of the surrounding environment [22,32,33], while discussions on the separate impacts of landscape patterns and topographic features are lacking. Furthermore, changes in water quality over years are usually not fully matched to the landscape pattern indicators for the corresponding year.

This research was mainly performed in Tangshan City, located in the center of Bohai Bay, which is the maritime hub of North China [34]. The shortage of water resources has restricted the sustainable development of the area [35]. Although water environment treatment projects implemented in recent years have improved the regional water quality, long-term integrated water management is still needed. In contrast to previous studies, this study calculated landscape metrics based on continuous years of land cover data (2018–2020) and corresponding to water quality parameters in relevant years. Focusing on the integrated effect of topography as well as landscape pattern on surface water quality, and employing redundancy analysis and variation partitioning, this study aimed to (1) evaluate multiple spatio-temporal variations in the role of landscape metrics in influencing river water quality changes, and (2) recognize respective contributions of topography and landscape patterns to changes in surface water environment. This study offered a foundation for finely assessing the response of regional inter-annual water quality to landscape patterns and improving the regional aquatic environment through rational land management that takes into account local topographic characteristics.

## 2. Materials and Methods

### 2.1. Study Area

The area of this study is mainly situated in Tangshan City, Hebei Province, China. Tangshan City (117°31′–119°19′E, 38°55′–40°28′N) is situated in the North China Plain and bordered by the Bohai Sea to the south (Figure 1). The main soil types are cinnamon soil and fluvo-aquic soil [36]. The topography is higher in the northern part of the area. The major land cover type of study area is cropland, and water area is primarily in the south. The climate is characterised by a typical warm temperate continental monsoon climate, and the annual mean temperature and rainfall are 11.5 °C and 607 mm, respectively [34]. The study area belongs to the Haihe River Basin, which is a typical water system in northern China and 80% of the annual precipitation is contributed by rainfall from June to September. Thus, the dry season is from October to May, and the wet season is from June to September.

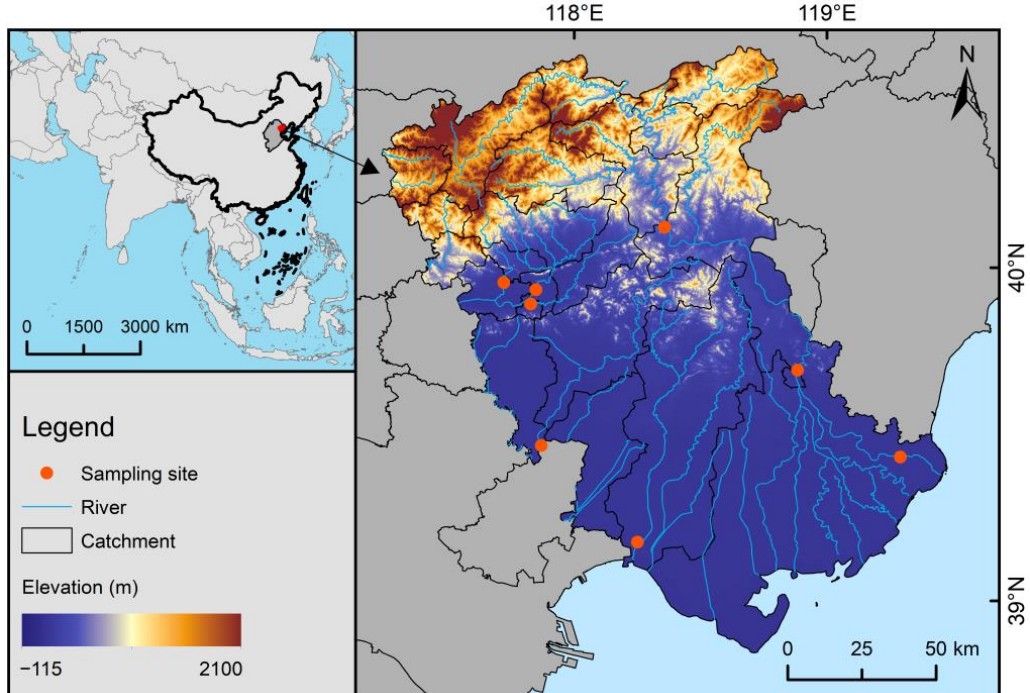

**Figure 1.** The geographical location of the study region. In the map showing the national boundaries, the dark grey area is the Haihe River basin in China and the red dotted area is the study area.

Tangshan City is one of the major industrial towns in northern China. The development of industry has increased regional water consumption [35]. Meanwhile, agriculture accounts for a comparatively high portion of water consumption [37]. The expansion of agriculture has further contributed to the shortage of water resources and the escalation of the conflict between water supply and demand.

### 2.2. Water Quality Monitoring and Analysis

This study set up a total of 8 sampling sites in the study region. The sampling data information is monthly data from January 2018 to December 2020, with data missing for the S1 site in July 2018, the S5 site in August 2020, the S6 site in February, April and May 2019, and the S8 site in December 2019 because of natural factors. Six water quality parameters were selected for analysis, including chemical oxygen demand ($COD_{Mn}$), dissolved oxygen (DO), total nitrogen (TN), ammonia nitrogen ($NH_4^+$-N), fluoride ($F^-$) and total phosphorus (TP). These parameters were determined according to the Environmental Quality Standards for Surface Water in China (GB 3838–2002).

## 2.3. Spatial Data and Landscape Metrics Data Acquisition and Analysis

The study was classified as buffer-scale group and catchment-scale group for the analysis of spatial scale effects. The radius of the circle buffer zone corresponding to each sampling point was 100 m, with the distance set with reference to previous research [38].

The landscape metrics consist of topographic feature metrics and landscape pattern metrics (Table S1). Considering relevant study [28], topographic feature metrics mainly reflecting regional surface relief and landscape pattern metrics indicating landscape composition and structure were selected. The topographic feature metrics included regional slope (Slope), regional relief (HD) and hypsometric integral (HI). The landscape pattern metrics included the percentage of each land cover type and the landscape structure at class level corresponding to each land cover type, such as patch density (PD), landscape shape index (LSI), and largest patch index (LPI). The catchment delineation information of the study region was obtained from the Chinese Academy of Environmental Planning. The ASTER GDEM V2 Digital Elevation Model (DEM) of the study region with 30 m accuracy was derived from the Geospatial Data Cloud site. Information on land cover for 2018, 2019 and 2020 in the study area with 30 m accuracy was obtained from [39]. The land cover was classified into seven categories containing cropland, forest, shrubland, grassland, water, barren land, and impervious layers (Figure S1). Shrubland and barren land were not included in the analysis because the area of shrubland and barren land accounted for less than 1%. The computing of topographic feature and landscape pattern metrics was based on DEM and land cover data, respectively. The construction of spatial scale and extraction of the land cover area were implemented using ArcGIS version 10.2 [40]. PD, LSI and LPI were computed using FRAGSTATS version 4.1 [41].

## 2.4. Statistical Analyses

The normality of the water quality parameters was examined using the Kolmogorov-Smirnov test. Mann-Whitney *U* test was conducted (except TN) to compare seasonal variations in water quality indicators because of the non-normality of the data. The Wilcoxon signed-rank test was used to distinguish the temporal differences in TN. Redundancy analysis (RDA) was applied to identify the relations between response variables (water quality indicators) and explanatory variables (landscape metrics). The best subset searching was applied to identify the best explanatory variables for water quality variations, and models that included variables with variance inflation factor > 10 (i.e., the models with high multicollinearity) were excluded [42]. A Monte Carlo permutation test ($n$ = 999) was performed to examine significance of the model and the constrained axes. The adjusted coefficients of determination of the models were employed to partition the variation in the response variables according to topographic features and landscape patterns [43]. The significance of portion of variance explained by each variable set in the variation partitioning was tested using the Monte Carlo permutation test ($n$ = 999) as well. Statistical analyses in the study were developed in R version 3.6.2 [44]. RDA analysis was conducted with the *vegan* [45] package.

## 3. Results

### 3.1. Seasonal and Spatial Differentiation of River Water Environment

The results of statistical analysis revealed water quality was less affected by seasons, except that DO and TN presented significant seasonal variations ($p < 0.001$). DO and TN presented higher concentrations in the dry season (Figure 2). Nevertheless, the concentrations of $COD_{Mn}$, $NH_4^+$-N, $F^-$ and TP were approximated in two seasons. Furthermore, there are some outliers for each water quality indicator during both seasons. Some of the outliers in the $COD_{Mn}$, TN and $NH_4^+$-N indicators exceed the limits of the Class IV standard of the Chinese Environmental Quality Standards for Surface Water, indicating that the water environment is polluted. Spatially, water quality indicators also presented spatial heterogeneity (Figure 3). The concentrations of water quality indicators in the west of the study region were generally high.

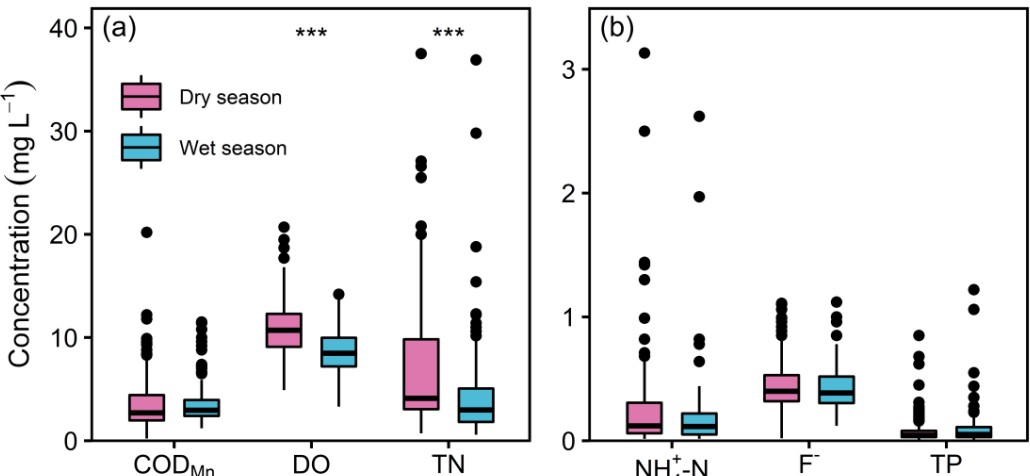

**Figure 2.** Water quality indicators during the dry and wet seasons. (**a**,**b**) The horizontal lines in the boxes indicate the median values, while the upper and lower border lines of the boxes indicate the upper quartile and lower quartile, respectively. *** indicates the statistical significance between the two seasons at $p < 0.001$. Abbreviations: $COD_{Mn}$, chemical oxygen demand; DO, dissolved oxygen; TN, total nitrogen; $NH_4^+$-N, ammonia nitrogen; $F^-$, fluoride; TP, total phosphorus.

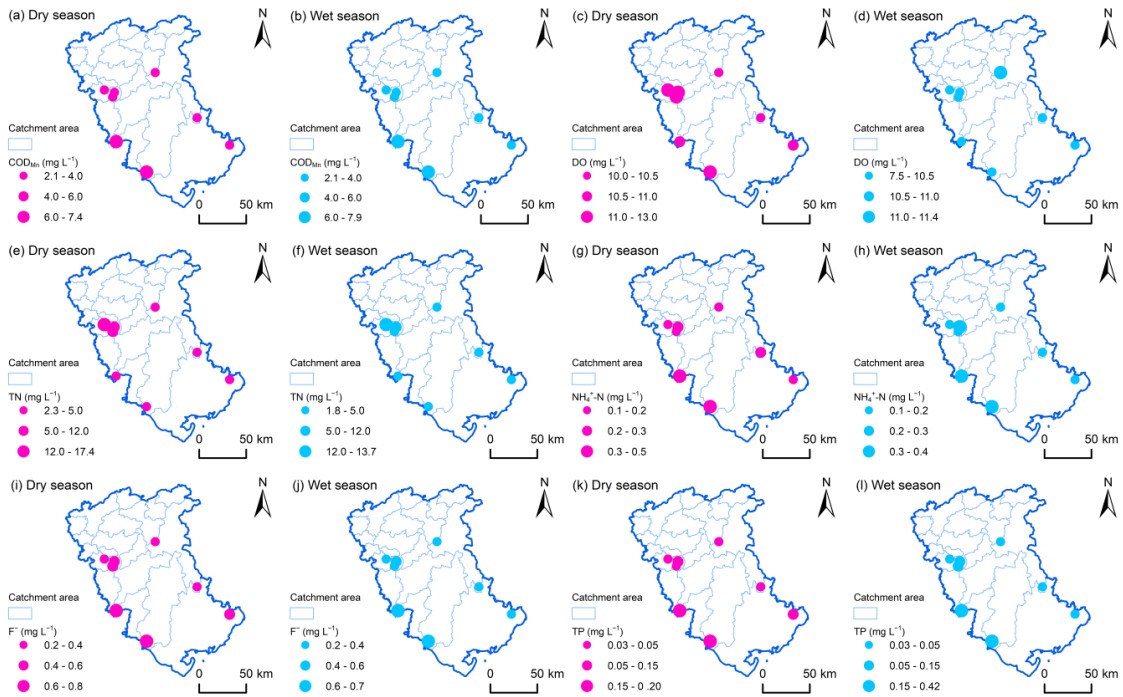

**Figure 3.** The spatial variation of $COD_{Mn}$ (**a**,**b**), DO (**c**,**d**), TN (**e**,**f**), $NH_4^+$-N (**g**,**h**), $F^-$ (**i**,**j**) and TP (**k**,**l**) during the two seasons.

### 3.2. Changes in Landscape Metrics at Multiple Scales

The mean values of Slope and HD at the catchment scale were higher than the buffer scale (Table S2). The mean value of HI was larger at the buffer scale. Similar to the indicators for topographic features, the landscape configuration and structure changed with the spatial scale. The dominant land cover types at the catchment scale were cropland and forest. Impervious layer and cropland together accounted for more than 75% at the buffer scale. On both spatial scales, grassland occupied the smallest proportion. At the buffer scale, the mean values of PD for all land cover types were higher over three years, and the mean values of LSI for each land cover type were higher at the catchment scale.

The proportion of cropland and impervious layer area increased from 2018 to 2020 at the catchment scale (Table S2). From 2018 to 2020, the proportion of forest, grassland and water area decreased by 2.09%, 0.11% and 0.26%, respectively. Similarly, the percentage of cropland increased over three years at the buffer scale. In 2018–2020, the proportion of grassland and water area decreased by 1.43% and 3.54%, respectively. Forest area accounted for 6.07% in 2018 and 2019, with the proportion dropping to 3.21% by 2020. The percentage of impervious layer area exceeded 50% over three years. At different spatial scales, the mean values of PD, LPI and LSI varied with landscape configuration changes over three years.

*3.3. Associations between Landscape Metrics and Aquatic Environment*

Redundancy analysis revealed the impact of landscape metrics on river water in relation to seasonal and spatial scales (Figure 4). At the catchment scale, HD, Forest, LSI$_{crop}$ and LSI$_{wat}$ were the best explanatory variables for water quality changes in the wet season, and the best explanatory variables for water quality changes in the dry season were HD, Grassland, PD$_{for}$, PD$_{imp}$, LPI$_{wat}$ and LSI$_{crop}$ (Figure 4a,b). The best explanatory variables explained approximately 53.3% and 71.8% of the variations in water quality during the wet and dry seasons, respectively. The interpretation rates for the first and second axes were 11.5% and 7.2% higher during the dry season than the wet season, respectively. The contribution of the first axis mainly originated from HD and LSI$_{crop}$ during the wet season, while PD$_{imp}$ made the largest contribution to the first axis during the dry season.

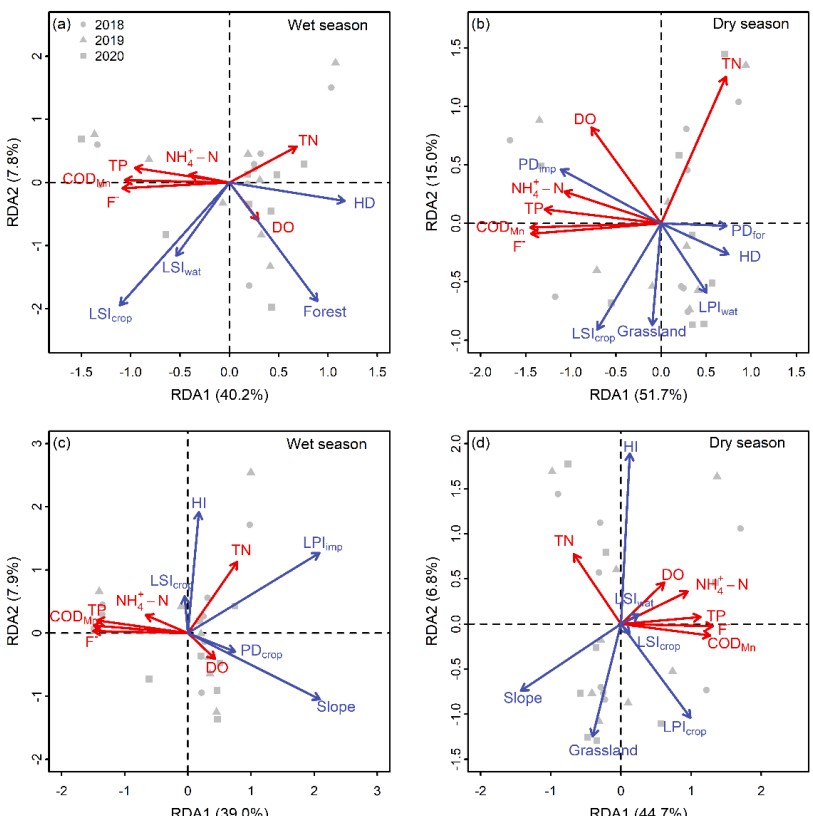

**Figure 4.** Ordination plots for the water quality variations at catchment (**a**,**b**) and buffer (**c**,**d**) scale in two seasons with landscape metrics as constraining variables. Abbreviations: COD$_{Mn}$, chemical oxygen demand; DO, dissolved oxygen; TN, total nitrogen; NH$_4^+$-N, ammonia nitrogen; F$^-$, fluoride; TP, total phosphorus; Forest and Grassland, the proportion of forest area and grassland area, respectively; Slope, regional slope; HD, regional relief; HI, hypsometric integral; PD$_{crop}$, PD$_{for}$, PD$_{imp}$, patch density of cropland, forest and impervious layer, respectively; LSI$_{crop}$ and LSI$_{wat}$, landscape shape index of cropland and water area, respectively; LPI$_{crop}$, LPI$_{wat}$, and LPI$_{imp}$, largest patch index of cropland, water area and impervious layer, respectively.

At the buffer scale, the analysis indicated that Slope, HI and $LSI_{crop}$ were the best explanatory variables for water quality variations in two seasons (Figure 4c,d). Meanwhile, the best explanatory variables in the wet season also include $PD_{crop}$ and $LPI_{imp}$, whereas Grassland, $LPI_{crop}$ and $LSI_{wat}$ comprised the best explanatory variables in the dry season as well. During the wet season, the best explanatory variables explained about 54.4% of changes in water quality. The first and second axes explained 39.0% and 7.9% of the information, respectively. During the dry season, 55.1% of the information on water quality changes could be explained by the best explanatory variables, of which 44.7% and 6.8% were interpreted by the first and second axes, respectively. Slope had a major contribution to the first axis in both seasons.

The distribution of points with different shapes was not significantly separated, indicating that there were no significant inter-annual variations in water quality. The permutation test confirmed that the models were significant ($p < 0.001$) for both two seasons at different spatial scales (Table S3). Moreover, the permutation test revealed that the first and second constraint axes of most models were significant. In the analysis of the dry season at the buffer zone, the first axis was significant ($p < 0.001$), and the second axis was not significant.

### 3.4. Variation Partitioning of Landscape Metrics

The variations of water quality associated with topographic features and landscape patterns were partitioned (Figure 5). At the catchment scale, the contribution of landscape patterns to variations of water quality was 36.7% and 72.6% higher than that of topographic features in dry and wet seasons (Figure 5a,b). The contribution of topographic features to river water variations at the buffer scale was greater than that at the catchment scale. Seasonally, landscape patterns explained more of the information on river water changes in the dry season. Topographic features and landscape patterns independently and significantly explained the water quality variations. Therefore, the jointly explained portion was negative.

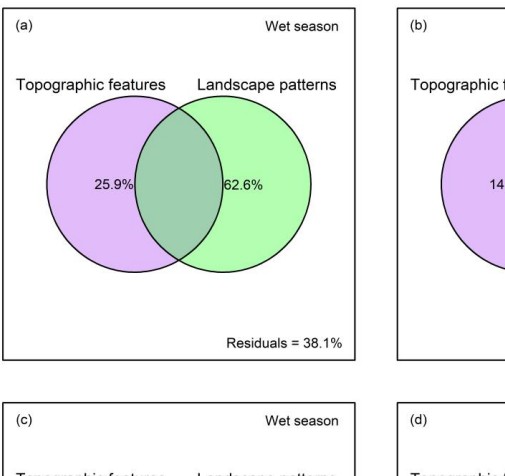
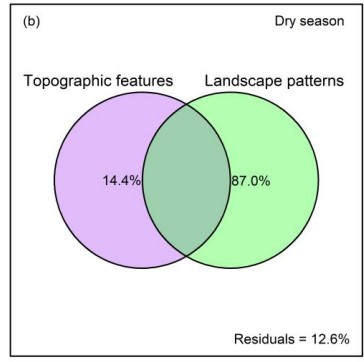
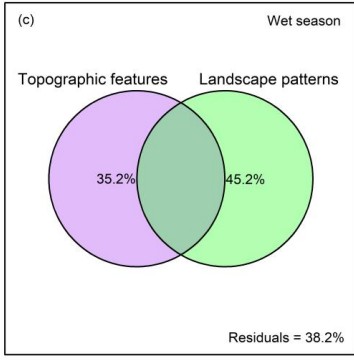
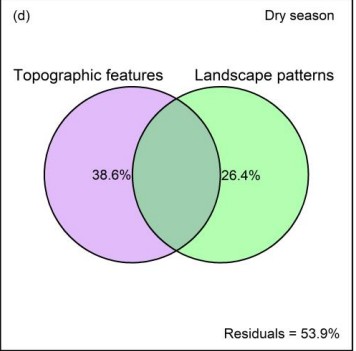

**Figure 5.** Diagrams describing the partitions of variation of water quality indicators by topographic features and landscape patterns at catchment (**a**,**b**) and buffer (**c**,**d**) scale in different seasons. Values less than zero are not displayed.

## 4. Discussion

### 4.1. Impacts of Topographic Features and Landscape Patterns on Water Environment

Consistent with the previous study [46], this research presented that water quality was strongly affected by topography. At the catchment scale, the indicator characterising the topographic relief of the corresponding area (HD) was the best explanatory variable for water quality changes in both seasons (Figure 4a,b). In addition, at the buffer scale, Slope and HI were the best explanatory variables for water quality changes (Figure 4c,d). This might be explained by the fact that topographic characteristics affect the number of pollutants entering the river by altering the velocity of surface runoff [47]. Especially, the slope is a basic indicator that can predict flow velocity of water flowing through the surface [48]. It is important to incorporate topographic feature parameters (e.g., slope) into the analysis of the complicated relations between landscape patterns and the aquatic environment.

This study presented that Forest and Grassland were negatively linked to TP and $NH_4^+$-N at the catchment (Figure 4a,b) and buffer scales (Figure 4d), indicating that forest and grassland had positive benefits on surface water. This impact stems mainly from the link between forest, grassland and NPS pollution, with reduced soil erosion and nutrient loss benefiting from favourable vegetation cover in forest and grassland [49–51]. On the contrary, the positive correlation presented by Forest and TN in the wet season might originate from the higher sediment yield into river bodies under stronger rainfall flushing [52]. Furthermore, the landscape pattern metrics of various land cover types, including metrics of landscape patch shape complexity and fragmentation, revealed a differential correlation with water quality parameters. This is related to the composition and configuration of diverse land cover types that influence the river water environment by modifying hydrological processes [22].

Overall, landscape patterns made a greater contribution to water quality variations than topographic features (Figure 5). The contribution of landscape patterns (26.4%) to water environment variability at the buffer scale was smaller than that of the topographic features (38.6%) in the dry season. Previous research suggested that landscape configuration metrics had higher impacts on the river water environment than physiographic and landscape composition metrics [28]. In this study, considering that the landscape structure was also divided according to land cover types, the percentage of land cover and landscape spatial configuration were unified to represent landscape patterns. Additionally, this study distinguished between the interpretation of topographic features or landscape patterns on water environment changes. Previous studies have revealed that topography could influence water quality by limiting land cover composition [53]. Therefore, topographic features and landscape patterns might be correlated and thereby overlap in the interpretation of water quality variations. However, the results of this study indicated that the role of topography and landscape patterns on water environment change was independent. This may be due to the variability of regional topographic gradients [54].

### 4.2. Impacts of Temporal and Spatial Scales

Similar to earlier works [55,56], this study indicated seasonal variations in the impacts of landscape metrics on aquatic environment variability. During the wet season, landscape pattern and topographic feature metrics explained an average of 53.8% of the variation in water quality at catchment and buffer scales (Table S3). In contrast, during the dry season, the average interpretation rate was 63.4%. This phenomenon might be related to seasonal water cycle changes [57]. Reduced precipitation during the dry season drives down river flow, decreases the dilution of pollutants and increases the impact of landscape on the aquatic environment [58]. Furthermore, the seasonal variability of the influence of landscape patterns on river water was higher than that of topographic features (Figure 5). This might be due to seasonal differences in surface runoff, which makes the landscape patterns more influential in pollutant interception and uptake.

Some findings have indicated that the impacts of landscape on water environment are closely linked to spatial scales [59,60]. Combining the results of wet and dry seasons in

this study, the best explanatory variables at the catchment scale provide higher information on water quality changes than those explained at the buffer scale (Table S3). Meanwhile, among the impacts of landscape composition and structure on the aquatic environment, the spatial scale was much stronger.

### 4.3. Limitations and Inspirations

This study concentrated on the spatio-temporal scale of topography and landscape pattern effects on water quality variability. This study used land cover data for three consecutive years (2018–2020) and calculated landscape pattern metrics for corresponding years, which contributed to matching the landscape pattern metrics to water quality in the corresponding years. The dataset is of high quality [39]. However, because of the impact of land reclamation in Bohai Bay, the area of impervious layers at low coastal elevations may be difficult to calculate very accurately and there is an acceptable error. It remains positive for understanding the impacts of land cover changes over time (under both natural and anthropogenic interactions) on the aquatic environment based on the inter-annual continuity of landscape information in this study.

RDA analysis has been widely applied to explore the relationships between landscape patterns and water quality [61,62]. Further research should analyse the differences in the influence of vegetation cover changes on water quality during wet and dry seasons, as vegetation can exert multiple effects on hydrology through physical and chemical processes [63]. It is necessary to integrate vegetation data from different seasons over years into the RDA analysis.

### 5. Conclusions

This study revealed that the response of river water quality to landscape metrics was affected by a combination of spatio-temporal dynamics and scale effects. During two seasons, the average interpretation of landscape patterns and topographic features on the variations of surface water quality at the catchment scale was better than at the buffer scale. Landscape connectivity and topographic relief might control the river water environment by altering the transmission of pollutants at the catchment scale. In turn, this effect might be attenuated within the buffer scale. The dominant contribution of landscape patterns and topographic features to the water quality varied with spatial and seasonal scales. These results further implied that optimal allocation of regional land resources could help to mitigate NPS pollution and thus improve the water environment.

**Supplementary Materials:** The following supporting information can be downloaded at: https://www.mdpi.com/article/10.3390/rs15051438/s1, Table S1: Descriptions of landscape metrics used in the study; Table S2: Summaries for landscape metrics from 2018 to 2020 at catchment and buffer scales; Table S3: The explained variation proportion of redundancy analysis (RDA) for the water quality with landscape metrics as constraining variables; Figure S1: Land cover of the study region in 2018 (a), 2019 (b), and 2020 (c).

**Author Contributions:** Conceptualization, L.D. and W.L.; methodology, W.L.; software, L.D. and W.L.; validation, X.L., Y.W. and L.W.; formal analysis, W.L.; investigation, L.D. and W.L.; resources, X.L.; data curation, L.D. and W.L.; writing—original draft preparation, L.D. and W.L.; writing—review and editing, X.L.; visualization, L.D. and W.L.; supervision, X.L.; project administration, L.D.; funding acquisition, L.W. All authors have read and agreed to the published version of the manuscript.

**Funding:** This research was supported by the Qinghai Province Key Research and Development and Transformation Program (2023-QY-205), the Basic Research Program of Qinghai Province (2023-ZJ-910M) and Jiangxi Province Key Research and Development Project (20212BBG73017).

**Data Availability Statement:** Data available on reasonable request.

**Conflicts of Interest:** The authors declare no conflict of interest.

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
