# Peer review of "Landscape Patterns and Topographic Features Affect Seasonal River Water Quality at Catchment and Buffer Scales"

_remotesensing, doi:10.3390/rs15051438_

Round 1
Reviewer 1 Report
Landscape patterns and topographic features affect seasonal river water quality at catchment and buffer scales:
· This manuscript does not have MDPI format.
· Add/Replace the name of the study area with the Keywords.
· In the last paragraph of the Introduction, the authors should mention the weak point of former works (identification of the gaps) and describe the novelties of the current investigation to justify that the paper deserves to be published in this journal.
· “Overall, the proportions of water quality variations explained by topographic features and landscape patterns at the catchment scale were high, with small proportions of residuals.”. Explain.
· In the Tables, highlight values that are more important and discuss them for better understanding readers.
· Focus on the advantages/disadvantages of the proposed method concerning the obtained results.
· At the end of the manuscript, explain the implications and future works considering the outputs of the current study.
· The quality of the language needs to be improved for grammatical style and word use.
Author Response
Point 1: This manuscript does not have MDPI format.
Response 1: Thank you. We have revised the formatting of sections such as the references.
Point 2: Add/Replace the name of the study area with the Keywords.
Response 2: Thanks for your suggestions. We have modified the text in the Keywords. The keywords modified or added are Seasonal river water quality and Haihe River Basin.
Point 3: In the last paragraph of the Introduction, the authors should mention the weak point of former works (identification of the gaps) and describe the novelties of the current investigation to justify that the paper deserves to be published in this journal.
Response 3: In response to your suggestions, we have revised and added a description of the shortcomings of previous studies in the penultimate paragraph of the introduction, as well as describing the focus of further research that should be undertaken (Line 68-75). The improvements and gap-filling aspects of this study compared to previous studies have been revised in the last paragraph of the introduction (Line 81-83). The revised text is as follows.
Revised text:
Line 68-75: Considering several spatial and temporal scales, there is no unified conclusion on the interpretation of the impact of landscape patterns and topography on the aquatic environment. Meanwhile, most studies focus on the total contributions or the contribution of a single landscape pattern indicator to water quality or integrity of the surrounding environment [22,32,33], while discussions on the separate impacts of landscape patterns and topographic features are lacking. Furthermore, changes in water quality over years are usually not fully matched to the landscape pattern indicators for the corresponding year.
Line 81-83: In contrast to previous studies, this study calculated landscape metrics based on continuous years of land cover data (2018-2020) and corresponding to water quality parameters in relevant years. Focusing on the integrated effect of topography, as well as landscape pattern on surface water quality...
Point 4: “Overall, the proportions of water quality variations explained by topographic features and landscape patterns at the catchment scale were high, with small proportions of residuals.”. Explain.
Response 4: We have revised the 3.4 section as a whole to make it clearer and easier to understand. We have also added seasonal comparisons to enrich the manuscript. The revised text is as follows (Line 248-256).
Revised text:
Line 248-256: The variations of water quality associated with topographic features and landscape patterns were partitioned (Fig. 5). At the catchment scale, the contribution of landscape patterns to variations of water quality was 36.7% and 72.6% higher than that of topographic features in dry and wet seasons (Fig. 5a and b). The contribution of topographic features to river water variations at the buffer scale was greater than that at the catchment scale. Seasonally, landscape patterns explained more of the information on river water changes in the dry season. Topographic features and landscape patterns independently and significantly explained the water quality variations. Therefore, the jointly explained portion was negative.
Point 5: In the Tables, highlight values that are more important and discuss them for better understanding readers.
Response 5: Thank you for your advice. We have put tables in the attachment. In addition, we have interpreted the relevant results in Table S2 in the manuscript (Line 200-207). Further, we have also added relevant discussion for the information in Table S3 (Line 305-308).
Revised text:
Line 305-308: During the wet season, landscape pattern and topographic feature metrics explained an average of 53.8% of the variation in water quality at catchment and buffer scales (Table S3). In contrast, during the dry season, the average interpretation rate was 63.4%.
Point 6: Focus on the advantages/disadvantages of the proposed method concerning the obtained results.
Response 6: Thank you. Further discussion has been provided about the methods used in the manuscript. This includes the breadth of application of the methods and the improvements and enhancements needed in future research. The revised text is as follows (Line 333-338).
Revised text:
Line 333-338: RDA analysis has been widely applied to explore the relationships between landscape patterns and water quality [61,62]. Further research should analyse the differences in the influence of vegetation cover changes on water quality during wet and dry seasons, as vegetation can exert multiple effects on hydrology through physical and chemical processes [63]. It is necessary to integrate vegetation data from different seasons over years into the RDA analysis.
Point 7: At the end of the manuscript, explain the implications and future works considering the outputs of the current study.
Response 7: We have added the main implications of this study and the improvements needed in the future. The revised text is as follows (Line 330-332, 334-338)
Revised text:
Line 330-332: It remains positive for understanding the impacts of land cover changes over time (under both natural and anthropogenic interactions) on the aquatic environment, as the inter-annual continuity of landscape information in this study.
Line 334-338: Further research should analyse the differences in the influence of vegetation cover changes on water quality during wet and dry seasons, as vegetation can exert multiple effects on hydrology through physical and chemical processes [63]. It is necessary to integrate vegetation data from different seasons over years into the RDA analysis.
Point 8: The quality of the language needs to be improved for grammatical style and word use.
Response 8: The language of the manuscript has been revised and embellished. Meanwhile, the revised manuscript has been proofread by all authors to improve the readability and avoid possible grammatical errors.
Again, many thanks for the reviews.
Best regards.
The authors

Reviewer 2 Report
While the work is of good qulaity, the writing needs to go through an extensive English editing. Many statements are hard to understand and/or follow along. English editing will highlight the effort made on this research and will make the work more appealing to readers. Attached comments are just an example.

Author Response
Point 1: While the work is of good qulaity, the writing needs to go through an extensive English editing. Many statements are hard to understand and/or follow along. English editing will highlight the effort made on this research and will make the work more appealing to readers. Attached comments are just an example.
Response 1: Thank you. The language of the manuscript has been revised and embellished. Meanwhile, the revised manuscript has been proofread by all authors to improve the readability and avoid possible grammatical errors. The changes to comments are as follows (Line 13-24, 30, 36-38). In addition, we have made other modifications to the manuscript.
Revised text:
Line 13-24: Redundancy analysis revealed that landscape patterns and topographic features have different impacts on the aquatic environment as seasonal spans and spatial scales change. Overall, the best explanatory variables explained an average of 58.6% of the variation in water quality at various spatial scales over the two seasons. Topographic features made a greater contribution to river water quality changes at the buffer scale; conversely, at the catchment scale, water quality changes stemmed primarily from differences in landscape composition and configuration. The landscape shape index of cropland (LSIcrop) was an important factor influencing seasonal river water quality changes at various spatial scales. These results suggest that considering landscape connectivity at distinct spatial scales could enhance the understanding of the alteration of hydrological processes across multiple topographic features, which in turn has an impact on seasonal river water.
Line 30: Deterioration of water quality is a worldwide problem...
Line 36-38: In recent years, the implementation of environmental policies has alleviated water pollution. Normally, the apparent spatio‐temporal variation in water quality is mainly caused by rainfall, soil erosion, climate perturbation and anthropogenic disturbance.
Again, many thanks for the reviews.
Best regards.
The authors

Reviewer 3 Report
Although this is an interesting study, it could be useful for understanding how the landscape patterns and topographic features affect seasonal river water quality in a watershed. However, the article needs some revisions. Please address the comments to improve the quality of your article.
1. Please add more references to your introduction (lines 59 to 65). I recommend that the authors should provide the following references: (a) Sliva, Lucie, and D. Dudley Williams. "Buffer zone versus whole catchment approaches to studying land use impact on river water quality." Water research 35, no. 14 (2001): 3462-3472
(b) Gao et al. (2022), Analyzing the critical locations in response of constructed and planned dams on the Mekong River Basin for environmental integrity, Environmental Research Communications, https://iopscience.iop.org/article/10.1088/2515-7620/ac9459.
2. Figures 1 require revision. On their study area map, authors are permitted to include river networks and systems. Please refer to the previously cited source.
3. Figures 2 and 4 need revision. Please review the manual for ArcGIS or another professional software in order to generate publication-quality figures.
4. Figures 3, 5 and 6 require revision. Authors may use Python or Matlab to generate publishable figures.
5. What is this research's significance? Please describe the potential implications of this study in a separate section (prior to the conclusion).
Author Response
Point 1: Please add more references to your introduction (lines 59 to 65). I recommend that the authors should provide the following references: (a) Sliva, Lucie, and D. Dudley Williams. "Buffer zone versus whole catchment approaches to studying land use impact on river water quality." Water research 35, no. 14 (2001): 3462-3472
(b) Gao et al. (2022), Analyzing the critical locations in response of constructed and planned dams on the Mekong River Basin for environmental integrity, Environmental Research Communications, https://iopscience.iop.org/article/10.1088/2515-7620/ac9459.
Response 1: Thank you. We have added your recommended article by Gao et al. (2022) in Line 72. In the section you mentioned, we have modified and added references (Line 70-75). The first reference you recommended we have cited in Line 55, reference 55 .
Revised text:
Line 70-75: Meanwhile, most studies focus on the total contributions or the contribution of a single landscape pattern indicator to water quality or integrity of the surrounding environment [22,32,33], while discussions on the separate impacts of landscape patterns and topographic features are lacking. Furthermore, changes in water quality over years are usually not fully matched to the landscape pattern indicators for the corresponding year.
Point 2: Figures 1 require revision. On their study area map, authors are permitted to include river networks and systems. Please refer to the previously cited source.
Response 2: Thank you for your advice. Considering the area of the region and the overall layout of points, we have not added more river networks, but we have also improved Figure 1 (Line 112). For example, beautify the colour of the picture and add the contour of the surrounding area.
Revised figure:

Point 3: Figures 2 and 4 need revision. Please review the manual for ArcGIS or another professional software in order to generate publication-quality figures.
Response 3: Thank you for your advice. We have put figure 2 in the attachment. With reference to other literature and based on the size of the overall picture, we have not added more information to the figure. Perhaps the current picture can reflect the information needed.
Point 4: Figures 3, 5 and 6 require revision. Authors may use Python or Matlab to generate publishable figures.
Response 4: Thank you for your valuable advice. In these figures, we have used the R for graphing, which has been widely used for statistical analysis and graphing in a variety of disciplines (https://www.R-project.org/). Some of the graphs have been retouched and adjusted.
Revised figures:
Figure 2 (Original figure 3):

Figure 5 (Original figure 6):

Point 5: What is this research's significance? Please describe the potential implications of this study in a separate section (prior to the conclusion).
Response 5: Thank you. We have included the significance and potential implications of the study separately in section 4.3. The revised text is as follows (Lin 324-332).
Revised text:
Line 324-332: This study used land cover data for three consecutive years (2018-2020) and calculated landscape pattern metrics for corresponding years, which contributed to matching the landscape pattern metrics to water quality in the corresponding years. The dataset is of high quality [39]. However, because of the impact of land reclamation in Bohai Bay, the area of impervious layer at low coastal elevations may be difficult to calculate very accurately and there is an acceptable error. It remains positive for understanding the impacts of land cover changes over time (under both natural and anthropogenic interactions) on the aquatic environment, as the inter-annual continuity of landscape information in this study.
Again, many thanks for the reviews.
Best regards.
The authors

Reviewer 4 Report
Suggest author to reduce the total percentage of the references that is more than 5 years.
Author Response
Point 1: Suggest author to reduce the total percentage of the references that is more than 5 years.
Response 1: Thank you for your suggestions. We have added and removed some references and have focused on adding references from the last five years. Some of the references added are as follows.
Some additional literature:
Cheng, X., Song, J., Yan, J., 2023. Influences of landscape pattern on water quality at multiple scales in an agricultural basin of western China. Environ. Pollut. 319, 120986. https://doi.org/10.1016/j.envpol.2022.120986
Gao, Y., Sarker, S., Sarker, T., Leta, O.T., 2022. Analyzing the critical locations in response of constructed and planned dams on the Mekong River Basin for environmental integrity. Environ. Res. Commun. 4, 101001. https://doi.org/10.1088/2515-7620/ac9459
Urgeghe, A.M., Mayor, Á.G., Turrión, D., Rodríguez, F., Bautista, S., 2021. Disentangling the independent effects of vegetation cover and pattern on runoff and sediment yield in dryland systems – Uncovering processes through mimicked plant patches. J. Arid Environ. 193, 104585. https://doi.org/10.1016/j.jaridenv.2021.104585
Xu, G., Cheng, Y., Zhao, C., Mao, J., Li, Z., Jia, L., Zhang, Y., Wang, B., 2023. Effects of driving factors at multi-spatial scales on seasonal runoff and sediment changes. CATENA 222, 106867. https://doi.org/10.1016/j.catena.2022.106867
Again, many thanks for the reviews.
Best regards.
The authors

Round 2
Reviewer 1 Report
Acceptable in the current form.
Reviewer 3 Report
Thanks for the revision.